# Gastrointestinal and Intra-Abdominal Mucormycosis in Non-Haematological Patients—A Comprehensive Review

**DOI:** 10.3390/jof11040298

**Published:** 2025-04-09

**Authors:** Benoît Henry, Alain Lefevre Utile, Stephane Jaureguiberry, Adela Angoulvant

**Affiliations:** 1Service des Maladies Infectieuses et Tropicales, APHP, Hôpital Universitaire de Bicêtre, 94275 Le Kremlin-Bicêtre, France; 2Service of Paediatrics, Department Women-Mother-Child, Lausanne University Hospital, 1005 Lausanne, Switzerland; 3Centre de Recherche en Epidémiologie et Santé des Populations (CESP), U1018, INSERM, 94807 Villejuif, France; 4Faculty of Medicine, University of Paris Saclay, AP-HP, 94275 Le Kremlin-Bicêtre, France

**Keywords:** mucormycosis, gastrointestinal, intra-abdominal, non-haematological, non-neonate, review

## Abstract

Intra-abdominal and gastrointestinal mucormycosis are less frequent than rhino-orbito-cerebral and pulmonary mucormycosis, but highly lethal. Their diagnosis remains challenging due to the non-specific clinical presentation. We collected English-language cases of intra-abdominal and gastrointestinal mucormycosis in non-haematological and non-neonatal patients published up to October 2024. This review analysed the epidemiological, clinical, and therapeutic charts of 290 cases. A proportion of 53.4% were reported from India and the USA. The main predisposing conditions were diabetes, solid organ transplant, ICU, and corticosteroid treatment. The most common site was the stomach (53.8%). Gastrointestinal perforation, skin breakdown, and abdominal wall infection were sources of intra-abdominal localisation. The most common symptoms were abdominal pain, vomiting, and gastrointestinal bleeding. The diagnosis relied on histology (93.8%), mycology with microscopy and culture (38.8%), and molecular methods (9.9%). Mortality (52.9%) was lower when treatment was intravenous amphotericin B, combined or not with surgery. Prompt treatment, essential for a favourable outcome, relies on early suspicion and diagnosis. Gastrointestinal and intra-abdominal mucormycosis should also be suspected in patients admitted in ICU with ventilation/nasogastric tube and corticosteroids and those with abdominal trauma or surgery, presenting abdominal distension, pain, and GI bleeding. Mycological diagnosis including direct examination, culture and Mucorales qPCR on tissue should assist with rapid diagnosis and thus treatment.

## 1. Introduction

Mucormycoses are life-threatening invasive infections caused by fungi of the order *Mucorales*, which until the last decade were known as Zygomycetes. Based on molecular phylogenetic analysis, taxonomy has evolved, the phylum *Zygomycota* has been abandoned, and these opportunistic human pathogens have been placed in the order Mucorales within the families Cunninghamellaceae, Lichtheimiaceae, Mucoraceae, Saksenaceae, and Syncephalastraceae [1,2]. Of more than 30 pathogenic species, those belonging to the genera *Rhizopus*, *Mucor*, and *Lichtheimia* (formerly *Absidia*) are reported to be responsible for most human mucormycoses [2,3]. Their identification has often been based on phenotypic characteristics, which are currently considered to be less accurate than molecular identification.

The worldwide increase in the number of cases during the last decades and the outbreak of mucormycosis in the context of the COVID-19 pandemic led the WHO to designate Mucorales as priority pathogens [4,5,6,7,8,9,10]. Infection usually occurs in debilitated patients such as those with haematological malignancies and bone marrow transplantation, solid organ transplantation (SOT), iron overload, diabetes mellitus, and the use of corticosteroids [3]. However, there have been reports of mucormycosis in immunocompetent hosts with trauma or burns over the last three decades [11,12,13,14,15,16].

Pulmonary, rhinocerebral, sinonasal, sino-orbital, and cutaneous localisations are the most common, accounting for more than 70% of cases. Gastrointestinal (GI) localisation is the least common, estimated at 5–13% of mucormycosis cases, and other intra-abdominal (IA) localisations are even rarer [3,17,18,19]. Their diagnosis remains a challenge due to the non-specific clinical presentation, especially when they occur outside the usual contexts such as those with haematological malignancies, neutropenia, SOT, and neonates [20,21].

We report here a comprehensive review of human mucormycoses with GI and IA localisations in patients without haematological malignancies and outside the neonatal context.

## 2. Patients and Methods

### 2.1. Search

We reviewed IA and GI mucormycoses cases from PubMed according to the PRISMA (Preferred Reporting Items for Systematic reviews and Meta-Analyses) guidelines for systematic reviews [22].

Each of the following key words were used: human, mucormycosis, zygomycosis, phycomycosis, abdominal, gastric, intestinal, colic, peritoneal, splenic, hepatic, disseminated, *Rhizopus*, *Mucor*, *Lichtheimia*, *Absidia*, *Rhizomucor*, *Apophysomyces*, *Saksenaea*, *Syncephalastrum*, *Cunninghamella*, and *Mucorales.* Renal and urinary tract infections (UTIs) without abdominal or peritoneal cavities involvement were not included in this review. Lists of articles in CSV (comma-separated values) format were generated and saved from PubMed, and then converted to Excel format.

The search was completed by reviewing the reference sections of original articles, of reviews, case series including those with pooled analysis and case reports and adding those that were not present in the PubMed CSV and summary files. The same inclusion criteria were used as for the PubMed search.

The inclusion criteria for this review were as follows: (1) articles in the English language; (2) absence of neonatal context and/or haematological disease and/or documented neutropenia; (3) documentation of proven *Mucorales* infection either histologically or by culture, or by molecular methods according to the European Organisation for Research and Treatment of Cancer (EORTC) and the Mycoses Study Group Education and Research Consortium (MSGERC) criteria for invasive fungal disease [23]; (4) cases with documentation of age and sex of the patient, underlying diseases/predisposing factors, site(s) of infection, methods of diagnosis, treatment, and outcome. Poorly described cases, including those from reviews and pooled cases, were excluded.

Data of sex, age, underling conditions, infection site, fungal agent, concomitant other infections, diagnostic methods (cytopathology, microscopic direct examination, cultures, and molecular methods), and antifungal and surgical treatment and outcome at 3 months after diagnosis were collected in an Excel table.

### 2.2. Definitions

GI and IA mucormycoses were defined as proven infections according to the EORTC and MSGERC criteria for invasive fungal diseases [23], the sites of infection being oesophagus, stomach, and small and large bowel and, respectively, abdominal wall, peritoneum, omentum, mesenteron, liver, spleen, and adrenal gland.

### 2.3. Statistics

A descriptive analysis of data and maps was performed using Microsoft Excel^®^. Categorical variables were expressed as the number of cases and as a percentage of the total (%). Comparisons between groups were made using Fisher’s exact test for continuous variables. *p* values < 0.05 were considered statistically significant.

## 3. Results

Our literature search of PubMed yielded a total of 363 articles. After excluding duplicate, non-English, and irrelevant articles, a total of 142 studies reported individual patient data. Together with the studies retrieved from the additional search of reference sections, 116 cases from 108 articles for a total of 290 cases were included (Appendix A). A flowchart of the sorting process is shown in Figure 1.

The first documented case was published in 1960. The chronological increase in the number of cases since the 2000s is illustrated in Figure 2.

Among the included studies, the countries reporting the highest number of GI/IA mucormycosis were India (93/290, 32.06%), USA (62/290, 21.37%), South Africa (19/290, 6.55%), Spain (14/290, 4.83%), and China (13/290, 4.48%) (Figure 3, Appendix A).

### 3.1. Demographic Data and Underling Conditions

Age information was available for 288 cases. The median age of included GI/IA mucormycosis patients was 45 years [1.5 month–86 years] and the majority of them were men (192/288, 66.67%). Thirty cases were reported in patients under 18 years, and 20 patients have age under 16 years.

Data of underling disease or condition were available for 281 cases. Their frequency varied across the decades and the countries (Table 1 and Appendix A). A majority of cases (220/281, 78.01%) had two or more underling diseases/conditions. Diabetes was the most common comorbidity (70/282, 24.82%), with no significant difference between men (39/70, 55.71%) and women (31/70, 44.29%). Of the patients with diabetes, 14.8% presented with ketoacidosis (KA).

SOT (46/282, 16.31%) with kidney and liver transplant was the most frequent (20/282, 7.09% and 15/282, 5.31%, respectively), and renal failure (36/282, 12.76%) were the most frequent comorbidities after diabetes. Abdominal surgery was reported in 42/282 cases (14.89%), with 37.5% (18/48) being performed for with GI perforation or peritonitis.

Cases associated with COVID-19 accounted for 8.51% of all cases and for 18.46% (24/130) of cases since 2020. Of the 24 cases, 14 (58.33%) occurred in India.

Intensive care unit (ICU) stay (67/282, 23.75%) and corticosteroid treatment (45/282, 15.96%) other than that prescribed in the context of SOT were common conditions in GI/AI mucormycosis cases.

Underlying conditions varied according to the GI and IA localisation (Figure 4). GI localisations had varied underling conditions. On the contrary, IA abscesses and abdominal wall infection abdominal injury and surgery were the main, whereas ICU stay was not reported.

### 3.2. Clinical and Paraclinical Presentation

The clinical and paraclinical characteristics of the cases (adult and paediatric) are shown in Appendix A. Of the 290 cases, 4 had no more precise information on the site than “GI” or “digestive”. In 59 cases, more than one GI and/or IA site was reported, and in 24 cases, the mucormycosis was disseminated.

Overall, stomach (156/290, 53.79%) was the most common site, followed by intestine (78/290, 26.89%), abdominal wall (26/290, 8.93%), and peritoneum/retroperitoneum (24/290, 8.24%) (Figure 5). The distribution according to patient age was different in patients younger than 16 years, with the intestine (11/20, 55%) being the most common site, followed by the stomach (5/20, 25%), liver (3/20, 15%), and peritoneum (2/20, 10%).

The large intestine (cecum, colon, rectum, anal canal) was most commonly involved (31/78, 39.74%), followed by the small intestine (duodenum, ileum, jejunum) (17/78, 21.79%) or both.

According to publication date, the distribution of sites differs after 2000, with fewer cases of disseminated mucormycosis and more cases of stomach, intestine, and GI/IA with more than one site (Figure 6).

### 3.3. Clinical Presentation

Symptoms were specified for 249 cases. They were non-specific with abdominal pain or discomfort (100/249, 40.16%) being the most common, followed by vomiting/nausea (51/249, 20.48%), GI bleeding (47/249, 18.87%), and fever (45/249, 18.07%) (Figure 7). All symptoms are summarised in Table 2.

Endoscopy was reported in 127/291 cases of mucormycosis: 93 were gastric, 4 were gastric with another intra-abdominal site other than the intestine, 6 were gastric and intestinal, 16 were intestinal, and 6 were disseminated with gastric or intestinal localisation.

Ulcer or erosion (96, 75.6%) and necrosis (44, 34.64%) were the main endoscopic features in both gastric and intestinal lesions. These were followed by the presence of exudate, mainly green or greyish (21, 16.53%), bleeding (20, 15.74%), infiltration or proliferative lesion (19, 14.96%), and mucosal inflammation or gastritis (13, 10.23%). Necrosis was reported more frequently in gastric localisation (34.64%), whereas mucosal inflammation or congestion was reported more frequently in intestinal localisation (Figure 8).

CT scan findings were reported in 114 cases and were not specific. Their characteristics are summarised in Appendix A.

### 3.4. Diagnostic

Diagnostic methods and results are summarised in Appendix A. In 272/290 patients, cytopathology was performed on biopsy, on tissue specimen obtained during surgery and on the contents of IA collection or abscess, showing large, irregular, non-septate hyphae. In 87 cases, cytopathology was performed only on biopsy, in 112 cases only on tissue sample and in one case only on abscess content. The diagnosis was made ante-mortem in one case and post-mortem in 45 (16.36%) cases, of which 38 were autopsies and 7 post-mortem biopsies.

A basic mycological diagnosis which included microscopy and/or fungal culture was reported in 104/271 (38.37%) cases: 30 with direct examination (DE) and culture, 73 with culture only, and one with DE only. In total, 30 cases were positive by DE and 90 by culture. Of these, 77 were GI or IA specimens, 6 were GI or IA and other sites, and 7 were other sites (2 peritoneal catheter tips, 1 peritoneal dialysis bag, 1 blood culture, 1 lung specimen, 1 sputum, and 1 skin). Of the 82 cases with a positive culture from a GI and/or IA specimen, 25 were from the stomach, 22 from the abdominal wall, and 16 from the peritoneum. In 84 cases, both cytopathology and mycology were performed. In 17 cases, the diagnosis was based on mycology because the cytopathology was negative, and in 14 cases, the diagnosis was based on cytopathology because the mycology was negative.

Molecular diagnosis, based on PCR and sequencing of rDNA regions, was performed in 27 cases: 15 on tissue samples, mostly on formalin-fixed and paraffin-embedded tissue (FFPET), and 12 on fungal culture for accurate identification.

Genus and species identification were available for 94/290 (32.41%) cases: 39 cases with genus identification and 53 cases with species identification, one case where cytopathology with immunochemistry could not differentiate between *Rhizomucor* sp. and *Rhizopus* sp. and one case where the molecular method on FFPET could not differentiate between *R. oryzae* and *R. sexualis* (Table 3).

*Rhizopus*, *Mucor*, *Apophysomyces*, and *Lichtheimia* were the most commonly isolated genera, with differences depending on the site of infection (Figure 9). *Saksanea vasiformis* and *Apophysomyces* were more frequently isolated from abdominal wall infections.

Bacterial (*Acinetobacter* sp., *Bacteroides fragilis*, *Clostridium difficile*, *Enterobacter* spp., *Enterococcus* spp., *Escerichia coli*, *Klebsiella pneumoniae*, *Pseudomonas aeruginosa*, *Stenotrophomonas maltophila*, *Serratia marcescens*, *Satphylococcus* spp., *Streptococcus viridians*, *Legionella pneumophila, Mycobacterium tuberculosis* complex), viral (SARS-CoV2, HIV, H1N1 influaenzae, CMV, HVA, HVB, HVC), fungal (*Candida* spp., *Aspergillus* spp., *Pneumocistis jirovecii*) or parasitic (*Strongyloides stercoralis*, *Schistosoma* sp., *Taenia solium* cysticercosis, *Plasmodium falciparum*, *Toxoplasma gondii*) co-infections were reported in 75/290 (25.6%) of the patients. They are summarised in Appendix A.

### 3.5. Management and Outcome

The management was documented in 285 out of 290 cases (Appendix A). In 40 out of 286 cases (29 diagnosed post-mortem), no treatment was recorded. The treatment modalities included antifungal therapy (AFT) (69/246, 28.16%), surgical interventions (41/246, 16.67%), and a combination of surgery and AFT (127/246, 51.62%). In eight cases, the intervention was limited to the removal of the peritoneal catheter and in two cases, to suturing the gastric ulcer.

Of the 203/290 (70%) cases for which AFT was reported, 3 did not specify the molecule. Amphotericin B (Amp B) intravenous derivatives were utilised as a first-line therapy, with Amp B deoxycholate (86/200, 43%), followed by the lipid formulations liposomal Amp B (79/200, 39.5%) and Amp B lipid complex (6/200, 3%). High-dose (≥5 mg/kg/d) administration of an Amp B lipid formulation was utilised in 34 cases. AmpB administered either as a monotherapy or in combination with other antifungal agents. In one case (abdominal wall fasciitis), Amp B deoxycholate was employed as a topical agent. Azoles, such as pozaconazole (PSZ) and isavuconazole (ISV), were predominantly utilised as maintenance therapy. PSZ was used as a first-line therapy (4/201, 2%), in combination (2/201, 1%), in combination and maintenance (3/201, 1.47%), and maintenance (28/201, 13.93%) therapy. Similarly, ISV was used as first line (2/201, 1%), in combination (2/201, 1%), and maintenance (3/201, 1.47%) therapy. Itraconazole (ITZ) was utilised as maintenance therapy in 3/201 (1.47%) cases, while fluconazole (FCZ) and ketoconazole (KTZ) were employed as a first-line therapy in 3 and 1 cases, respectively. Micafungin and caspofungin were used in combination in 7/201 (2.48%) and 2/201 (1.47%) cases, respectively. In conjunction with the administration of AFT and/or surgical intervention, a reduction in immunosuppression was reported in 16/39 (41.02%) patients with SOT for whom treatment was specified.

Outcome information at 90 days was available for 272 cases, and the 90-day mortality rate was 52.94% (see Appendix A). This mortality rate differed significantly (*p* < 0.01) according to the treatment regimen, with the highest mortality rate (83.78%) being observed in patients treated only with surgery, in comparison to all other main regimens (see Figure 10).

No significant differences were observed among the other treatment regimens, despite the lower mortality rate observed in subjects treated with Amb lipid formulations.

The mortality rate exhibited variation across different countries (Figure 3) and according to the infection sites (Figure 11). No significant differences were observed based on age, with a median age of 45 years for both survivors (0.5 y–82 y) and non-survivors (0.13 y–86 y) and between adults (52.89%) and children (53.33%).

## 4. Discussion

This comprehensive review of GI and/or IA mucormycosis in non-neonatal and non-haematological contexts collected data on the epidemiology, diagnosis, management, and outcome of 290 cases between 1960 and 2024.

It is not possible to estimate the incidence or prevalence of GI and IA mucormycosis, but in the present study, the number of reported confirmed cases increased after 2010. This may reflect either a true increase in GI and/or IA mucormycosis due to increased awareness and improved diagnostic methods, or more reports of this fungal infection. For example, the outbreak of mucormycosis associated with the COVID-19 pandemic, particularly in India, led to the WHO issuing a warning on this infection. The overall prevalence of mucormycosis in India was estimated to be almost 70 times higher than that in the global data [9]. However, in this study, the prevalence of GI and IA mucormycosis in children and adults, without haematological disease and outside the neonatal context, in India was less than twice that of the USA, which ranked second in case numbers. This may be due to a lower number of cases of GI/IA infections in this population compared to India, as GI cases were reported mainly in premature neonates prior to 1990 [21], but also to the overall similar underlying diseases and conditions.

Diabetes mellitus, SOT, especially renal, and renal failure were the most common underlying diseases of GI and IA mucormycosis, with frequencies that may vary between countries (Appendix A). Among countries with more than 10 reported cases, the percentage of cases with diabetes and renal failure (5%) was lower in South Africa, where abdominal trauma predominated (26%), and that of SOT was higher in the USA (21%) and China (23%). Cases with HIV were most commonly reported in the USA (10%) and Spain (14%) and those with COVID-19 (15%) in India and China. The percentage of cases admitted to intensive care and treated with corticosteroids was higher in Spain (64% and 71%, respectively) and lower in South Africa (5% for both). Gastric and intestinal mucosa and abdominal wall injuries, either disease-related or iatrogenic, were potential portals of entry for *Mucorales*. As for the other clinical forms, ICU admission, abdominal surgery and corticosteroid treatment were common underlying conditions, suggesting that they may be healthcare-associated risk factors for GI and IA mucormycosis [24].

The source of GI and IA mucormycosis remains unclear, but the ingestion or introduction of Mucorales spores with contaminated food, drink, or medical devices is strongly suspected as a route of entry through the digestive tract. On the other hand, in cases with abdominal wall involvement, the main route of entry appears to be through the skin from environmental sources.

As in other studies, the stomach was the most commonly affected GI site, followed by the colon and ileum, with an inverse distribution in children under 16 years of age [21], and then liver, spleen, and IA cavity following.

As neither clinical nor CTS or USG features are specific, the diagnosis of GI mucormycosis remains a challenge with 17% of cases diagnosed post or perimortem, particularly before 2000. Abdominal distension and pain and GI bleeding as haematemesis, haematochezia or melena were the common features for GI forms or swelling, fasciitis, necrosis for IA forms with abdominal wall as the primary site. Ulceration or erosion was the most common but non-specific endoscopic GI feature. However, necrosis, reported in more than 30% of cases and reflecting vascular invasion, should be an alerting feature.

Improving the diagnostic delay is therefore important in GI ulcerations, IA abscesses and abdominal wall fasciitis, even if the patient is immunocompetent, the differential diagnosis should include mucormycosis. Microscopic examination of wet mounts with KOH or fluorescent brighteners such as calcofluor white (Sigma Aldrich, St. Louis, MO, USA) or blankophor (Tanatax Chemicals, Ede, The Netherlands) is one of the cornerstones in the diagnosis of mucormycosis [25] and strongly recommended since it provides rapid at least presumptive diagnosis [26]. However, in this study, it was only performed in 11.76% of the cases. Direct examination needs to be confirmed by histopathology for evidence of tissue invasion of non-sterile sites and by culture and/or molecular or in situ identification techniques when such assays are available [26]. Overall, fungal cultures are positive in 50% to 71% of cases of mucormycosis [17,25]. In this study, culture was only performed in 38% of cases and was positive in 74.04% of cases and similar to that reported by Didehdar et al. (14/19, 73.68%) for 87 cases reported between 2015 and 2021 [27]. This high rate may be due to publication bias.

The identification of the fungal agent was mainly based on phenotypic characteristics. Molecular methods, mainly based on PCR and the sequencing of the ITS1 and ITS2 regions of the rDNA, have been performed in 27 cases since 2003, of which 24 allowed the identification of Mucorales. Gastroenterologists and surgeons should be encouraged to send samples for mycological examination to determine the exact aetiological agent.

Worldwide, more than 90% of mucormycoses are caused by *Rhizopus* spp., *Mucor* spp. and *Lichtheimia* (formerly *Absidia*) spp. In contrast, *Cunninghamella* spp., *Apophysomyces* spp., *Saksenaea* spp., *Rhizomucor* spp., *Cokeromyces* spp., and *Syncephalastrum* each account for fewer than 5% of cases. However, there are geographical variations, with *Mucor* spp. and *Lichtheimia* spp. as secondary causes in the Americas and Europe, and *Apophysomyces* spp. as secondary causes in India [3]. The genera *Rhizopus*, *Mucor*, *Apophysomyces*, and *Lichtheimia* were identified with varying frequencies according to the site of infection (Figure 9). The genus *Apophysomyces*, with a particular emphasis on *A. elegans*, was the third most isolated genus. This is likely attributable to the inclusion of IA mucormycosis with abdominal wall start, predominantly from India, for which this genus was identified in 9/13 cases. However, it is important to note that the identification of these genera relied predominantly on phenotypic characteristics (70/94, 74.46%), which are currently considered less accurate than molecular identification. Consequently, the interpretation of genera and species epidemiology should be approached with a degree of caution.

The guidelines for the treatment of mucormycosis recommend a combination of surgical debridement and AFT with a high dose of Amp B lipid preparations of as a first-line therapy [26]. However, in settings where resources are limited, Amp B deoxycholate may be the only option. In the present review, 51.41% of the treated patients received the combined therapy with surgery and Amp B lipid preparations (39.80%) or with Amp B deoxycholate (42.78%). High-dose (≥5 mg/kg/d) of Amp B lipid formulations was used in 39.53% of cases. Since 2010, there have been reports of the utilisation of PSZ or ISZ, primarily in the context of maintenance or combination therapy.

In the present review, crude mortality at three months was 52.94%, which is comparable with that of cases with only one GI or IA localisation (40% to 50%) and lower than that of cases with more than one localisation (57%) or with disseminated infection (82%). This is consistent with previous publications, which reported a mortality between 40 and 85%, with lower mortality reported after 2010 [17,21,27,28,29]. The mortality rates of adults (52.89%) and children (53.33%) were comparable. This can be attributed to the exclusion of the neonatal population, in which the mortality rate of GI mucormycosis is significantly higher, ranging from 70% to 85% [27,29]. Abdominal wall mucormycosis has been documented in various case reports and reviews, predominantly concerning cutaneous localisations, with an overall mortality rate of 35% [30]. Abdominal wall mucormycosis has been documented in various case reports and reviews, predominantly concerning cutaneous localisations, with an overall mortality rate of 35%. Thus, specific mortality of abdominal wall mucormycosis with deep subcutaneous involvement (beyond the subcutaneous plane) was not reported. This review found that the mortality rate for cases involving the abdominal wall was higher (59.09%), with 53.33% of cases occurring as the sole site of infection and 71.43% of cases involving other sites.

The mortality rate exhibited a significant variation according to the treatment regimen, with the highest mortality rate (83.78%) observed in patients treated exclusively with surgery, in comparison to all other main regimens and a lower mortality rate found in subjects treated with Amb lipid formulations with or without surgery. The low mortality (25%) observed in patients treated with only Amb lipid formulations is intriguing. It is noteworthy that 12/15 (80%) of cases exhibited solely gastric involvement, and vascular invasion was documented in only 3 of these cases, suggesting a low degree of invasion as a potential contributing factor to the favourable outcomes observed. Furthermore, half of the survivors received prolonged AFT (2 to 6 months) with liposomal Amb or liposomal Amb followed by PSZ. This experience is limited, but it suggests that when surgery is not possible, prolonged AFT may be an option.

The present review is subject to certain limitations. Firstly, it was retrospective in nature and primarily consists of case reports and case series, thus being limited by the absence of prospective and controlled trials. Even if it has been systematic, some cases may have been overlooked. Secondly, publication bias must be noted, since cases considered unusual are more likely to be reported or published. However, the analysed data relied on stringent inclusion criteria.

## 5. Conclusions

GI and IA mucormycosis are a deadly fungal infection, the diagnosis of which is complicated by its rarity and non-specific clinical presentation. Consequently, these infections are under-suspected. It is therefore vital that suspicion of mucormycosis is raised at the earliest opportunity, and that treatment is commenced promptly, in order to ensure a favourable outcome [26]. In light of this, we propose to suspect GI and IA mucormycosis in patients at risk, as well as those admitted to the ICU with ventilation/nasogastric tube and corticosteroids, exhibiting symptoms such as abdominal distension and pain, and fever, particularly when associated with GI bleeding. The diagnosis of mucormycosis can be expedited through the utilisation of mycological diagnostic methods, encompassing direct examination and culture on invasive samples (endoscopic biopsy, surgical sample), and Mucorales qPCR performed on tissue and blood samples in select circumstances. These approaches facilitate a swift diagnosis of mucormycosis prior to histological examination. The treatment regimen should encompass AFT, predicated on Amp B derivatives, and surgical intervention when feasible.

## Figures and Tables

**Figure 1 jof-11-00298-f001:**
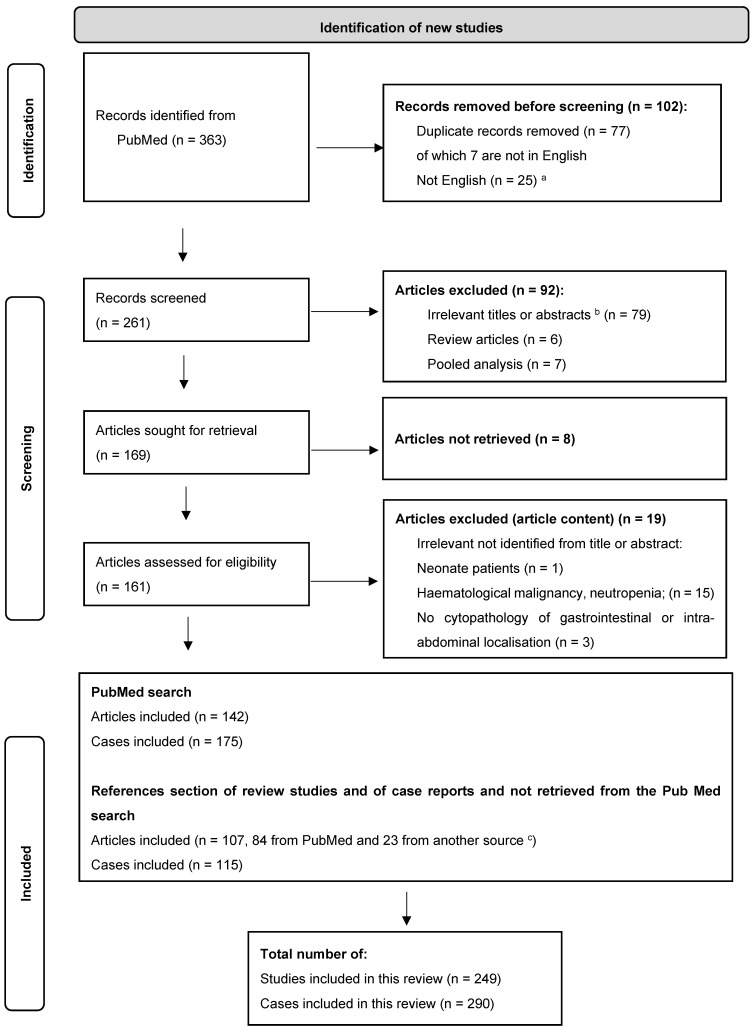
PRISMA 2020 flow diagram [22]. a. Of these, 9 articles reported cases which met the other inclusion criteria of gastrointestinal and intra-abdominal mucormycosis: Spanish language (*n* = 2: 1 gastric, 1 intestinal), French (*n* = 2: 1 hepatic, 1 disseminated with hepatic, splenic, and gastric involvement), German (1 gastro-duodenal), Italian (1 peritoneal), Czech (1 gastric), Norwegian (1 gastric), Chinese (1 gastric). b. Irrelevant title or abstract: not mucormycosis (e.g., blastomycosis); neonatal or haematological, including neutropenia, contexts; no intra-abdominal or digestive proven localisation. c. Other sources: ScienceDirect.com, Elsevier, journal site (e.g., The American Journal of Gastroenterology—ACG).

**Figure 2 jof-11-00298-f002:**
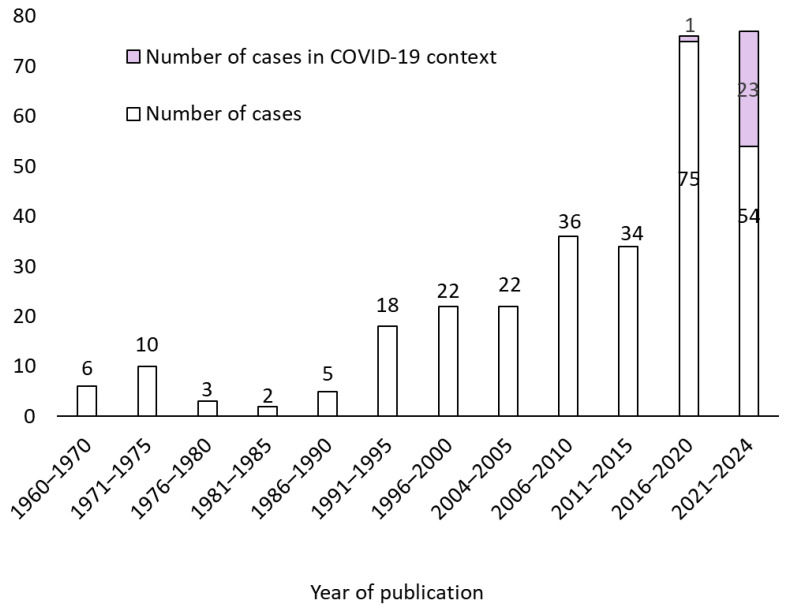
Number of published cases per year from 1960 to October 2024.

**Figure 3 jof-11-00298-f003:**
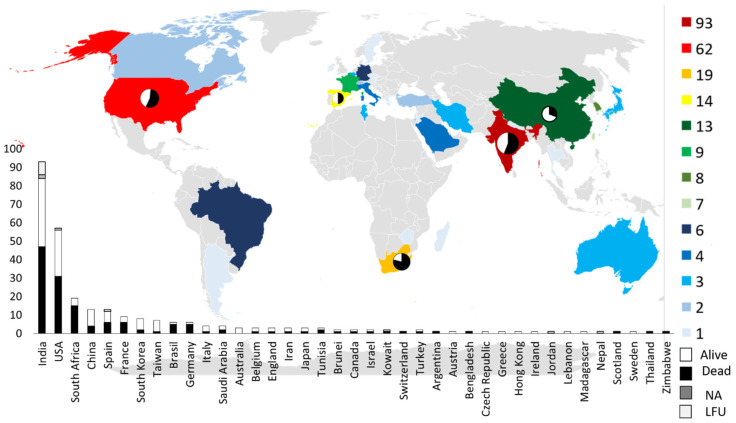
Geographical distribution of cases. NA, not available; LFU, lost to follow-up.

**Figure 4 jof-11-00298-f004:**
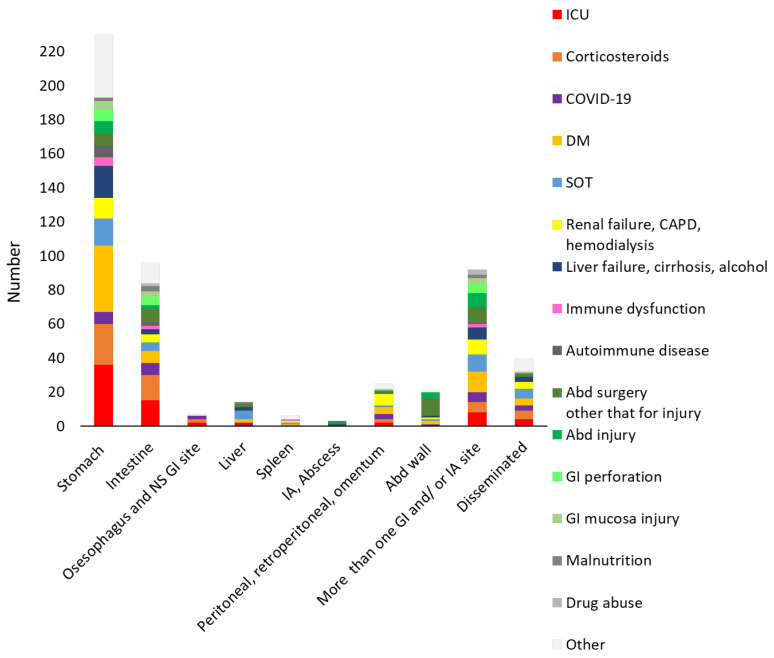
Underlying diseases and conditions distribution according to the gastrointestinal (GI) and intra-abdominal (IA) localisations. For each localisation, the sum of underlying diseases/conditions is greater than the number of each localisation because in most cases there was more than one. ICU, intensive care unit; DM, diabetes mellitus; SOT, solid organ transplant; CAPD, continuous ambulatory peritoneal dialysis; NS, not specified. “Other” includes the “other” category as summarised in Table 1.

**Figure 5 jof-11-00298-f005:**
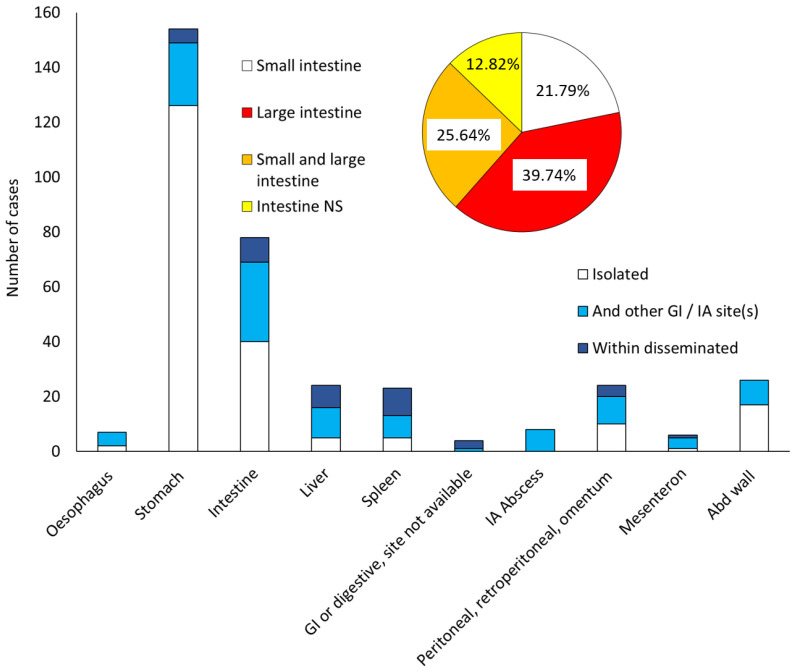
Gastrointestinal (GI) and intra-abdominal (IA) mucormycosis sites. Abd, abdominal; NS, not specified. Within disseminated: disseminated cases of mucormycosis with a GI and/or IA localisation.

**Figure 6 jof-11-00298-f006:**
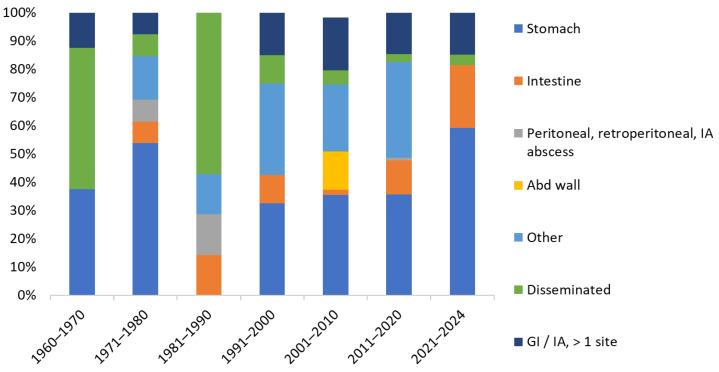
Site distribution (%) according to the publication period. Abd, abdominal.

**Figure 7 jof-11-00298-f007:**
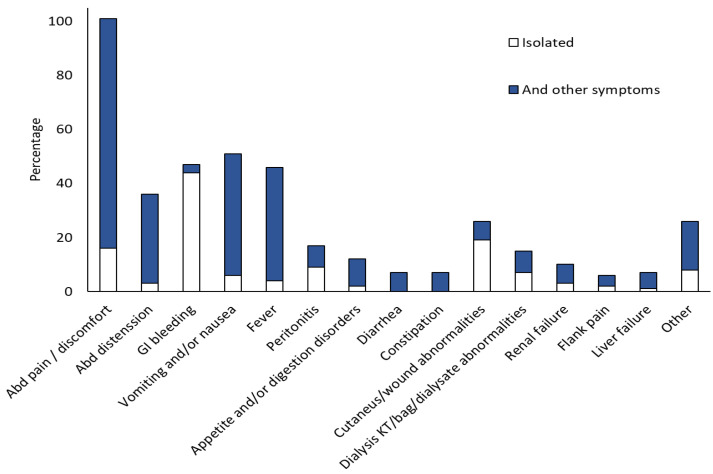
Symptoms rate (percentage). GI, gastrointestinal. Abd, abdominal. Dialysis catheter or bag or dialysate abnormalities (clogged Tenckhoff catheter, cloudy dialysis bag, or dialysate fluid). Other symptoms: acute abdomen, bowel movements, mass in epigastrium, hepato-splenomegaly, splenomegaly (HSM), paralytic ileus, pneumoperitoneum, rectum mucosal sloughing, anaemia, multi-organ failure (MOF), sepsis, and symptoms linked to other site in disseminated cases (cardiac tamponade, chest pain, dyspnoea, mental confusion).

**Figure 8 jof-11-00298-f008:**
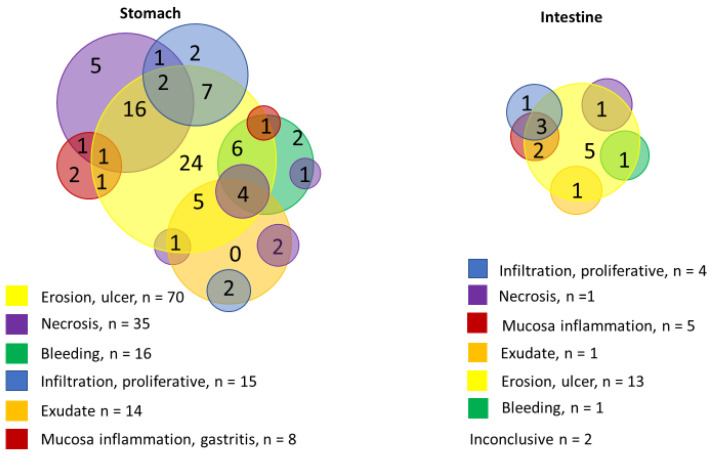
Endoscopic features distribution according to the gastric versus the intestinal localisation.

**Figure 9 jof-11-00298-f009:**
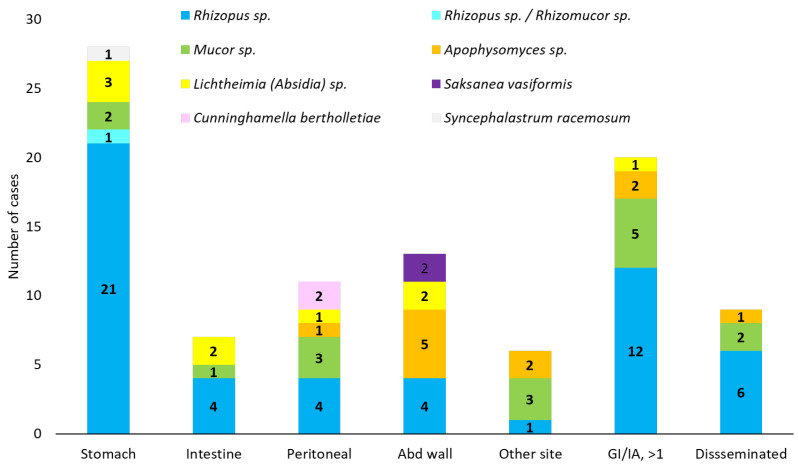
*Mucorales* genus distribution according to the gastrointestinal (GI) and intra-abdominal (IA) localisation. Abd, abdominal. GI/IA, >1, GI/IA localisation with more than 1 site: *Mucor* sp. (3 with stomach infection, 2 with peritoneal infection). Disseminated: *Rhizopus* sp., all cases were from USA and Europe.

**Figure 10 jof-11-00298-f010:**
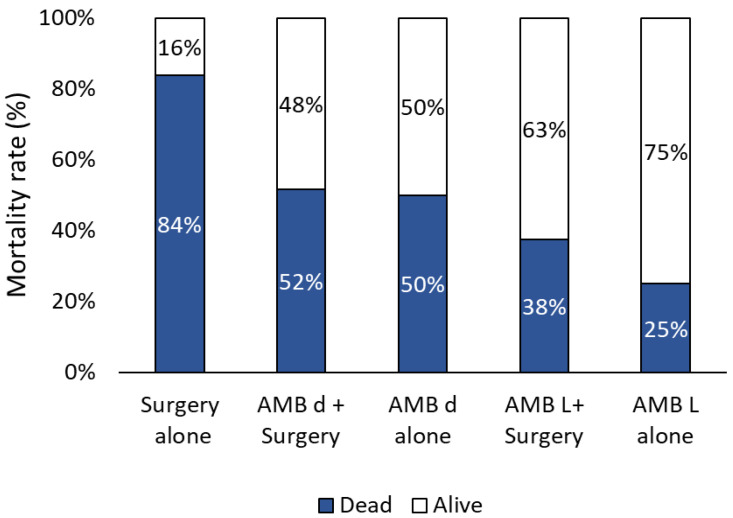
Outcome (%) according to the main treatment regimens. AMB d, amphotericin B deoxycholate. AMB L, amphotericin B lipid formulations.

**Figure 11 jof-11-00298-f011:**
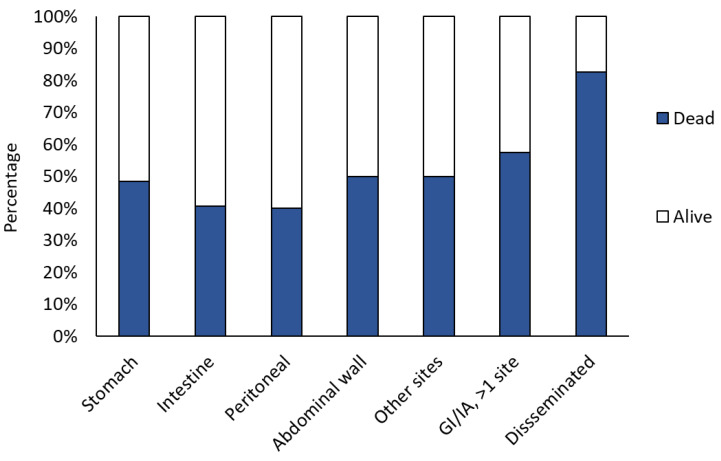
Outcome percentage according to the infection site. GI, gastrointestinal; IA, Intra-abdominal.

**Table 1 jof-11-00298-t001:** Underlying disease or condition. The sum of underlying diseases/conditions is greater than 291 (100%) because in most cases there was more than one. Percentage calculated only for number ≥ 5. CAPD, Continuous ambulatory peritoneal dialysis; HIV, human immunodeficiency virus; AIDS, acquired immune deficiency syndrome; ARDS, acute respiratory distress syndrome; ECMO, extra corporeal membrane oxygenation; ICU, intensive care unit. a. Metabolic acidosis does not include diabetic keto-acidosis. b. Other: (i) GI/IA, pelvic condition (*n* = 25): gastritis (*n* = 3), gastric erosion, adenocarcinoma, portal gastropathy, gastroparesis, peptic ulcer history, gastro-enteritis, *Salmonella* enteritis, *C. difficile* colitis, pan-colitis, digestive adenocarcinoma, diarrhoea, pancreatitis, peritoneal infection/peritonitis history (*n* = 6), breach in CAPD sterile technique, percutaneous drainage of renal collection, postpartum haemorrhage, testicular cancer + chemotherapy, sider bite (*n* = 1); (ii) thoracic/pulmonary condition (*n* = 18): ARDS (9), COPD (*n* = 3), respiratory failure, emphysema, H1N1 pneumonia, pulmonary tuberculosis, pleural infection, chest stab, thoracic surgery (*n* = 1); (iii) infection history (*n* = 4): typhoid fever (2), rhino-facial mucormycosis, *P. falciparum* malaria (*n* = 1); iron overload/transfusions (*n* = 5); (iiii) miscellaneous (*n* = 12): tobacco use (*n* = 5), burn (*n* = 2), back trauma, pesticide ingestion, cytopenia, haemolytic uremic syndrome, motor hemiplegia (*n* = 1).

Underlying Disease/Condition	Cases	
	Number	Percentage
Diabetes mellitus, of which with ketoacidosis (*n* = 10)	70 10	24.8 3.5
Solid organ transplantation Kidney only Liver only Other: kidney + liver (*n* = 4); heart (*n* = 3); heart + lung (*n* = 2), kidney + pancreas, lung, multi-organs (*n* = 1)	46 20 14 12	16.3 7.1 5 4.2
Autoimmune disease Inflammatory bowel disease (*n* = 4) Systemic lupus erythematosus (*n* = 3), Inflammatory vasculitis (*n* = 2), Sarcoidosis and rheumatic heart disease (*n* = 1)	11	3.9
Renal failure/dysfunction, of which 10 occurred in diabetic patients CAPD Haemodialysis, No replacement for kidney dysfunction (*n* = 4)	36 21 11	12.8 7.5 3.9
Hepatic failure/dysfunction, of which Cirrhosis	19 8	6.7 2.8
Impaired immune response: HIV infection, of which AIDS, CD4 count > 200/mm^3^ (*n* = 4) CD4 low count or dysfunction (*n* = 3) Splenic absence or splenectomy (*n* = 2) Other impaired immune response: Papillon–Lefevre syndrome, Down’s syndrome, low absolute lymphocyte count, and monocyte HLA-DR dysfunction (*n* = 1)	18 10 6	6.4 3.5 2.1
COVID-19	24	8.5
Alcohol abuse, of which hepatic failure/dysfunction (*n* = 2) and cirrhosis (*n* = 1)	23	8.1
Drug abuse	9	3.2
Malnutrition (none without other underlying condition)	5	1.8
Abdominal injury Traffic accident Blunt/stab wound Fall/professional accident (*n* = 4) Gunshot/bombing (*n* = 3)	22 9 6	7.8 3.2 2.1
Abdominal surgery, of which for gastrointestinal perforation/peritonitis	42 18	14.9 6.4
Stay in ICU, of which Septic shock/sepsis Metabolic acidosis ^a^ Multi-organ failure, Mechanical Ventilation ECMO	67 25 12 16 22 5	23.7 8.9 4.2 5.7 7.8 1.8
Corticosteroids (excluding TOS immunosuppression), of which auto-immune disease treatment (*n* = 5)	45	16
Other ^b^	50	17.7
None	6	2.1

**Table 2 jof-11-00298-t002:** Symptoms distribution for the 249 cases with available information. IA, intra-abdominal: peritoneal (*n* = 14), peritoneal + intra-abdominal abscess (*n* = 4) infection, intra-abdominal abscess. Abd. Wall, abdominal wall. GI, gastrointestinal. a. Dialysis catheter or bag or dialysate abnormalities (clogged Tenckhoff catheter, cloudy dialysis bag or dialysate fluid). b. Other symptoms: acute abdomen, bowel movements, mass in epigastrium, HSM, hepatosplenomegaly (*n* = 2), paralytic ileus, pneumoperitoneum (*n* = 2), rectum mucosal sloughing, anaemia, multi-organ failure (MOF), sepsis, and symptoms linked to other site in disseminated cases (cardiac tamponade, chest pain, dyspnoea, mental confusion). c. Other sites: spleen (*n* = 5), liver (*n* = 3), oesophagus (*n* = 1), mesenteron (*n* = 1).

	Stomach (*n* = 111)	Intestine (*n* = 33)	Peritoneal (*n* = 13)	Abd. Wall (*n* = 15)	Other Sites ^c^ (*n* = 10)	**GI/IA, >1 Site** **(*n* = 56)**	**Disseminated** **(*n* = 16)**
Abdominal pain/discomfort	59 (53.15%)	12 (36.36%)	5 (38.46%)	3 (20%)	2 (20%)	18 (32.14%)	1 (5.88%)
Abdominal distension	11 (9.91%)	13 (39.39%)	2 (15.38%)	0	0	8 (14.28%)	2 (11.76%)
GI bleeding	52 (46.85%)	12 (36.36%)	0	0	5 (50)	13 (23.21%)	2 (11.76%)
Vomiting and/or nausea	27 (24.32%)	13 (39.39%)	2 (15.38%)	1 (6.67%)	0	6 (10.71%)	2 (11.76%)
Fever	15 (13.51%)	3 (23.08%)	2 (15.38%)	6 (40%)	2 (20%)	1 (1.78%)	5 (29.41%)
Peritonitis	1 (6.67%)	6 (18.18%)	1 (7.69%)	0	0	6 (10.71%)	2 (11.76%)
Appetite/digestion disorders	10 (9.01%)	1 (3.03%)	0	0	1 (10%)	0	0
Diarrhoea	1 (0.90%)	2 (6.06%)	1 (7.69%)	0	0	2 (3,57%)	0
Constipation	2 (1.80%)	4 (12.12%)	0	0	0	1 (1.78%)	0
Cutaneous/wound abnormalities	0	2 (6.06%)	1 (7.69%)	14 (93.33%)	1 (10%)	8 (14.28%)	0
Dialysis catheter or bag or dialysate abnormalities ^a^	0	0	4 (30.77%)	0	0	4 (7.14%)	0
Renal failure	0	0	0	0	1 (10%)	7 (12.5%)	2 (11.76%)
Flank pain	0	0	0	0	2 (20%)	4 (7.14%)	0
Liver failure	2 (1.80%)	0	1 (7.69%)	0	0	2 (3.57%)	2 (11.76%)
Other ^b^	3 (2.70%)	1 (3.03%)	0	0	2 (20%)	5 (8.92%)	12 (70.59%)

**Table 3 jof-11-00298-t003:** Genus and species identification for the 95 cases for which it was reported.

Identification	No (%)
*Rhizopus*	52 (54.7)
* Rhizopus* sp.	23 (24.2)
* R. microsporus*	12 (12.6)
* R. microsporus var. rhizopodiformis*	6 (6.3)
* R. microsporus var. microsporus*	1 (1)
* R. microsporus var. azygosporus*	1 (1)
* R. oryzae (arrhizus)*	7 (7.4)
* R. oryzae* or *R. sexualis*	1 (1)
* R. stolonifer*	1 (1)
*Rhizomucor* sp./*Rhizopus* sp.	1 (1)
*Mucor*	16 (16.8)
* Mucor* sp.	13 (13.7)
* M. indicus*	2 (2.1)
* M. ramosissimus*	1 (1)
*Apophysomyces*	11 (11.6)
* A. elegans*	10 (10.5)
* A. variabilis*	1 (1)
*Lichtheimia* (former *Absidia*)	9 (9.5)
* Lichtheimia* (former *Absidia*) sp.	2 (2.1)
* L. corymbifera*	3 (3.6)
* L. ramosa*	3 (3.6)
* L. hongkongensis*	1 (1)
*Cunninghamella bertholletiae*	2 (2.1)
*Saksanea vasiformis*	2 (2.1)
*Syncephalastrum racemosum*	1 (1)

## Data Availability

All analysed data are included in this paper.

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
