# Peer review of "Gastrointestinal and Intra-Abdominal Mucormycosis in Non-Haematological Patients—A Comprehensive Review"

_jof, 2025, doi:10.3390/jof11040298_

Round 1
Reviewer 1 Report
The paper is a comprehensive review of gastrointestinal and intraabdominal mucormycosis. Overall the authors do a great job of collecting and collating the available data.
- Table 1 - this table has several areas which are difficult to understand. Specifically, "Solid organ transplantation" - should "Kidney" and "Liver" be "Kidney only" and "Liver only"? Should "hearth" be "heart"?
- Table 1 - "Renal failure/dysfunction" - rows 3 and 4 are unclear as written
- Figure 4 - unclear as presented - How do percentages add up to 100%? Where there no patients with more than one risk factor? Example - which risk factor group does one place a diabetic man with solid organ transplantation, who is on corticosteroids and is in renal failure in the ICU after Covid-19 infection, into?
- Table 2 - Please clarify the "symptom" "Dialysis KT or bag / dialysate"
- Figure 8 - As above, how do the percentages total 100? Did no patient have more than one finding on endoscopy? Example - where do you place the person with mucosal inflammation and bleeding ulcer?
- Table 3 - please check "Mucor" numbers
- Management and Outcome - please use "combination" rather than "additive", as additive denotes an unproven property of combining antifungal agents
Author Response
Please see the attachement

Reviewer 2 Report
The manuscript by Henry et al. provides an interesting review of a rare clinical entity (abdominal and gastrointestinal mucormycosis) which is of interest to the reader of Journal of Fungi. The strength of this review is the detailed information that is provided for a large number of cases (n=290).
- General comment. Display percentages with 1 decimal place
- Abstract line 25. Specify what is meant with ‘mycology methods’. Also molecular tests can be mycology methods.
- The criteria were defined by the EORTC and MSGERC (add MSGERC).
- Figure 4. ‘Risk factors distribution (%) according to the gastrointestinal (GI) and intra-abdominal (IA) localisatons. “Other” includes the “other” category as summarized in Table 1 and tobacco use’. In figure 4 underlying conditions are presented which does not equal risk factors. Why is tabacco used included?
- Figure 5. It is not clear what is meant with ‘within disseminated’
- Diagnosis Line 219. The difference between biopsy and tissue sample is not clear.
- Diagnosis lines 220-221. ‘The diagnosis was made ante-mortem in one case and post-mortem in 45 (16.36%) cases, of which 38 were autopsies’. This phrasing is confusing as there are 290 cases in total. 38/45 post-mortem diagnoses were autopsies. How were the other 7 post-mortem diagnosis made?
- Diagnosis line 222. It is unclear why direct examination is called ‘a mycological diagnosis’ and molecular diagnosis not.
Reviewer 3 Report
Dear authors, I have evaluated your article "Gastrointestinal and intra-abdominal mucormycosis in non-haematological patients, a comprehensive review". The review is written in good English and is easily read. The theme of your review presents interest to both scientific community and medical doctors and therefore is worth publishing. There are some minor imperfections that can be easily corrected. The whole text looks a bit inaccurate and should be revised. The main problem is the absence of the full name before you use abbreviations. I would recommend to carefully revise the article from this point of view so that the abbreviations would not appear without the full names and the full names would not be repeated before already explained abbreviations. I will mention some examples in the detailed comments section.
Line 74 - this is the first time the abbreviation CSV appears in the text. The full name (comma-separated values) should be present.
line 80 - you used only articles in English. It is not clear how many articles/cases were missed and what languages they belonged. It seems, you could include at least articles in French and German. Modern abilities of Google provide possibility to analyse articles in other languages.
line 199 - you should explain what is (MOF).
line 246 - Abd - does not have a full name.
Author Response
Please see the attachement"
